# Comparison of the Influence of Different Nucleic Acid Extraction Assays on the Sensitivity of *Trypanosoma cruzi*-Specific Real-Time PCR

**DOI:** 10.3390/microorganisms10081554

**Published:** 2022-07-31

**Authors:** Simone Kann, Wendy Zabala-Monterroza, Cenia García, Gustavo Concha, Olfert Landt, Andreas Hahn, Felix Weinreich, Hagen Frickmann

**Affiliations:** 1Medical Mission Institute, 97074 Würzburg, Germany; 2Public Health Laboratory, Secretariat of Health, Department Cesar, Valledupar 20000001, Colombia; wendyzabalam@gmail.com (W.Z.-M.); celigajurado@gmail.com (C.G.); 3Organization Wiwa Yugumaiun Bunkauanarrua Tayrona (OWYBT), Department Health Advocacy, Valledupar 2000001, Colombia; gustavoconcha16@gmail.com; 4TibMolBiol, 10103 Berlin, Germany; olandt@tib-molbiol.de; 5Institute for Medical Microbiology, Virology and Hygiene, University Medicine Rostock, 18057 Rostock, Germany; andreas.hahn@uni-rostock.de (A.H.); hagen.frickmann@med.uni-rostock.de (H.F.); 6Department of Microbiology and Hospital Hygiene, Bundeswehr Hospital Hamburg, 20359 Hamburg, Germany; felixweinreich@bundeswehr.org

**Keywords:** Chagas, pre-analytics, evaluation, diagnostic accuracy, test comparison

## Abstract

For the molecular diagnosis of Chagas disease by real-time PCR (polymerase chain reaction), optimization of diagnostic accuracy is desirable. The detection limit of real-time PCR assays for the diagnosis of *Trypanosoma cruzi* in human serum is affected by various influences including the choice of the nucleic acid extraction assay. In this study, three nucleic acid extraction assays were compared regarding their influence on the sensitivity of a *T. cruzi*-specific real-time PCR with 62 reference sera containing *T. cruzi* target DNA (deoxyribonucleotide acid). More than 95% of the positive sera were correctly identified after all three nucleic acid extraction strategies with a detection rate ranging from 96.8% (60/62) for the worst assay to 100% (62/62) for the best one. A matched pairs analysis for the comparison of the cycle threshold (Ct) values obtained with the 59 reference samples with positive real-time PCR results after all three nucleic acid extraction schemes indicated differences in a range of about 3 Ct steps. Summarized, all three compared nucleic acid extraction schemes were basically suitable for *T. cruzi*-specific PCR from serum with some minor differences. However, in the case of low quantities of circulating parasite DNA in the serum of a patient with Chagas disease, even minor effects can make a difference in the individual diagnosis.

## 1. Introduction

Poverty-related Chagas disease, which is, e.g., endemic in resource-limited tropical settings in South America [1], has been of increasing interest for the development of molecular diagnostic test assays in the recent two decades [2,3,4,5,6,7,8,9,10,11,12,13,14,15]. However, different from other systemic parasitic diseases such as malaria, for which excellent diagnostic accuracy of molecular diagnostic approaches has been repeatedly shown [16,17,18], the reliability of molecular techniques for the diagnosis of Chagas disease has often been less convincing in previous studies [8,10,11,12,13,14]. Reasons for the observed varying reliability of *Trypanosoma cruzi*-specific molecular diagnostic assays comprise the pathogen’s high genetic variability, but also its close phylogenetic relationship to other parasites [11,12,13,14]. As another factor, influencing the diagnostic accuracy with particular relevance for sensitivity, it has been described that the whole diagnostic process needs to be optimized, which also includes the choice of the most suitable nucleic acid extraction technique [8].

The relevance of adequate nucleic acid extraction approaches for the diagnostic reliability of the molecular diagnostic detection of parasitic diseases is not specific to Chagas disease only. For helminth infections, in particular, the diagnostic sensitivity of real-time PCR-based detection strongly depends on the choice of the applied extraction strategy [19,20,21]. Protocols comprising digestion steps and bead beating-based disruption of pathogen cells to release their target DNA have been described as particularly useful [19,20,21]. Even the appropriate choice of the beats for the bead beating has been shown to make a difference regarding the yield of target DNA within the sample [20]. Interestingly, the superiority of more robust extraction procedures compared to standard procedures was not found to be consistent for all helminth species [22].

Protozoan parasites are no exemption either. The reliability of their diagnostic detection with molecular assays has been shown to depend on the sufficient release of target DNA from parasite cells. Again, harsh extraction assays containing digestion, freeze-thawing, or bead-beating steps were associated with superior results compared to less robust extraction approaches [23,24].

In this study, three different nucleic acid extraction assays were applied with serum samples from Colombian Indigenous people with known active Chagas disease and subsequent analysis with the NDO (“newly developed one”) real-time PCR, a recently described assay specifically amplifying *T. cruzi* DNA from human serum samples [15]. By doing so, optimization of nucleic acid extraction with the aim to improve sensitivity for *T. cruzi* detection from human serum was intended.

## 2. Materials and Methods

### 2.1. Sample Materials

A total of 62 frozen residual serum sample materials from Indigenous people living in remote areas of Colombia with active Chagas disease were included in the assessment. Circulation of *T. cruzi* DNA in serum had been confirmed by PCR in a previously published study [15] as well as in the course of other yet unpublished investigations. More than this, the reference materials had been pre-characterized by multiple approaches as detailed elsewhere [15], including rapid antigen testing, serology based on enzyme-linked immunosorbent assay (ELISA) and/or immunofluorescence testing (IFT), real PCR for kinetoplast DNA (kDNA), the 18 S ribosomal ribonucleotide acid gene (18 S rRNA), and *T. cruzi* nuclear DNA (TCZ), in 9 out of 62 cases even by sequencing of PCR amplicons. So, Chagas disease was considered as well confirmed. The samples had been collected from Chagas patients at the Colombian sites Ashintukwa (n = 20), Ahuyamal (n = 6), Cherua (n = 2), Dungakare (n = 1), Marocaso (n = 3), Sabannah Crespo (n = 2), Sabannah de Higuieron (n = 5), Seminke (n = 14), and Tezhumake (n = 9). The patients’ female:male ratio was nearly balanced with 32 females and 30 males, the mean age ± standard deviation was 24.5 ± 17.5, and the median age (interquartile range IQR) was 19.5 (12; 34). Prior to further assessments, the serum samples were stored deep-frozen at −80 °C to preserve the quality of the DNA.

### 2.2. Applied Nucleic Acid Extraction Schemes

All included frozen serum samples were subjected to nucleic acid extraction as described by the manufacturers’ protocols, applying the following three nucleic acid extraction assays: the RTP (“Ready-to-Prep”) Pathogen kit (Invitek Molecular GmbH, Berlin, Germany; later referred to as RTP assay), the MagaBio plus Virus DNA/RNA purification kit version 2 (Hangzhou Bioer Technology Co., Ltd., Hangzhou, China; later referred to as the MagaBio assay), and the EZ1&2 Virus Mini Kit v2.0 performed on an EZ1 extraction automate (Qiagen, Hilden, Germany; later referred to as the Qiagen assay). The extracted sample volumes and the obtained eluate volumes were in a similar range over the compared extraction schemes with 200 µL and 60 µL for the RTP assay, 300 µL and 80 µL for the MagaBio assay, as well as 200 µL and 60 µL for the Qiagen assay, respectively. The extractions were performed in close temporal association to avoid bias due to different states of DNA degradation within the samples. The eluates were stored at −80 °C prior to further assessment by real-time PCR.

### 2.3. Applied Real-Time PCR Targeting T. cruzi DNA in the Eluates

The Chagas real-time PCR (NDO-PCR, patented, purchased at TibMolBiol, Berlin, Germany: *T. cruzi* LightMix^®^, Ref 53-0755-96, Phocid Herpes Virus (PhHV) Extraction control reference 66-0901-96, Lyophilized 1-step RT-(real-time-)PCR Polymerase Mix, Cat-No 90-9999-96) was applied with all nucleic acid eluates as described previously [15,25] on a RotorGene Q cycler (Qiagen, Hilden, Germany). As a minor modification compared to a previous description of the assay, the probe sequence had been slightly altered to 5′-TCG + AACCCC + ACCTCC-3′, the “+” symbol marks locked nucleic acid (LNA) bases included to alter the annealing temperature. Positive control samples containing target DNA, as well as PCR-grade water-based negative controls, were included in each run. Extraction, as well as sample inhibition, was controlled, applying the commercial Phocid herpes virus-(PhHV-)DNA-based PhHV spiked extraction control (DNA) assay mentioned above. To simulate routine-like diagnostic conditions, each eluate was only assessed once by the *T. cruzi*-specific PCR. Only samples with negative results in the *T. cruzi*-specific PCR were repeatedly amplified once to exclude random errors during pipetting of the PCR assays. A schematic representation of the diagnostic workflow is indicated in Figure 1.

### 2.4. Statistical Assessment

Due to the low number of included positive reference sample materials in the assessment, reaction failures were just descriptively recorded. Samples, for which a positive *T. cruzi*-specific PCR result was recorded from all three nucleic acid extraction attempts, were included in the comparison of the cycle threshold (Ct) values of the real-time PCR. After passing Kolmogorov–Smirnov testing for normality, the Ct values of the paired rows were subjected to repeated measures ANOVA (analysis of variance) with multi comparison post testing according to Tukey–Kramer in case of significant results in the ANOVA assessment. The analysis was performed applying the software GraphPad Instat, version 3.06 (GraphPad Software Inc., La Jolla, San Diego, CA, USA).

### 2.5. Ethics

Ethical clearances applicable for this study comprised the clearance “2019_HA 163 Acta No 2019-4”, provided by the Ethics Committee for Investigation, Bogota, Colombia in 2019, the clearance “2016_HA 190 Acta No 032018”, provided by the Ethics Committee, St. Marta, Colombia in 2018 and also the clearance “BWF/H/52228/2012/13.10.10-1/3.4,6Tropendiagnostik, Acto No 0022013”, provided by the Ethics committee Valledupar, Colombia, in 2013. Informed consent was obtained from the study participants or next to kin. In addition, ethical clearance for the technical evaluation was granted by the medical association of Hamburg, Germany (reference number: WF-011/19, obtained on 11 March 2019), allowing anonymous use of residual sample materials for test comparison purposes even without informed consent. The assessments were conducted according to the guidelines of the Declaration of Helsinki.

## 3. Results

### 3.1. Effects of the Nucleic Acid Extraction Schemes on PCR Failure in the Qualitative Assessment

Positive *T. cruzi*-specific PCR results with 100% (62/62) of the assessed reference materials from patients with active Chagas disease were recorded after nucleic acid extraction with the RTP assay only. The MagaBio assay scored second-best with 98.4% (61/62) correctly identified samples containing *T. cruzi*-specific DNA, closely followed by the Qiagen assay with 96.8% (60/62) correct identifications. All positive samples showed positive results already during the first run, while the failed reactions remained negative in the repetitions. The extraction and inhibition controls did not indicate sample inhibition as a source of interference.

### 3.2. Effects of the Nucleic Acid Extraction Schemes on the Recorded Cycle Threshold (Ct) Values

Due to positive *T. cruzi*-specific PCR results, 59 samples were included in the matched pair analysis of the measured cycle threshold (Ct) values. As indicated in Table 1, the lowest mean Ct values were calculated for the RTP assay followed by the Qiagen assay and the MagaBio assay, with standard deviations in a comparable range for all three assays (Table 1). The mean difference between the RTP extraction assay and the MagaBio assay was less than 3 Ct values and, thus, less than a decadic logarithmic step. The repeated measures ANOVA confirmed variation between the quantitative results of the three compared assays (*p* < 0.0001). In the Tukey–Kramer multiple comparisons test, the significance of lower Ct values in comparison to the MagaBio assay could be confirmed for the RTP assay (*p* < 0.001) and the Qiagen assay (*p* < 0.05), while no statistically significant difference between the RTP assay and the Qiagen assay was observed. Effective matching was confirmed by the assumption test with a *p*-value of 0.0078.

Focusing on the three reference materials with discordant results after real-time PCR from the eluates obtained with the different nucleic acid extraction assays, the sample missed after extraction with the MagaBio assay showed Ct values of 32.9 and 22.0 in the PCR reactions after extraction with the RTP assay and the Qiagen assay, respectively. For the two samples that were missed by PCR after the Qiagen assay-based extraction, Ct values of 32.2 and 33.5 were measured after extraction with the RTP assay, while Ct values of 36.2 and 38.4 were recorded after extraction with the MagaBio assay for the two samples missed by the Qiagen assay.

The raw data of the measured Ct values for all 62 samples subjected to all three compared nucleic acid extraction schemes are shown in Appendix A Table A1.

## 4. Discussion

The study was performed to assess the effects of three different nucleic acid extraction schemes on the diagnostic sensitivity of real-time PCR for the identification of *T. cruzi*-specific DNA in human serum samples. With all three applied nucleic acid extraction schemes, more than 95% of the assessed *T. cruzi* DNA-positive reference samples were correctly identified as positive. However, minor differences were observed and so, only extraction with the RTP assay was associated with the detection of all positive samples, while one sample was missed after extraction with the MagaBio assay and two after extraction with the Qiagen assay. Thereby, the reaction failures affected different samples for both types of nucleic acid extraction. Compared to the calculated mean values obtained with the eluates from the three compared nucleic acid extraction methods, the samples with the failed PCR reactions after individual extraction schemes showed mostly higher cycle threshold (Ct) values with the exemption of a very low Ct value after extraction with the Qiagen extraction for the sample missed after extraction with the MagaBio assay. These higher Ct values are indicative of low amounts of target DNA, explaining the individually failed reactions with target DNA amounts close to the diagnostic detection threshold. However, and as shown in a previously published study, which introduced the performed *T. cruzi*-specific PCR [15], target DNA amounts in serum samples of Chagas patients are often close to the detection threshold. While a previous study had shown good sensitivity of more than 90% [15] for the *T. cruzi*-specific PCR applied in the present investigation, individual samples had gone undetected due to low parasitemia close to the diagnostic detection threshold. Accordingly, even such minor differences as observed with the compared nucleic acid extraction assays may indeed be of relevance for the diagnosis in individual patients.

When focusing on Ct value-based semi-quantification rather than on qualitative PCR results alone, repeated measures ANOVA was performed in order to assess whether the eluates from the different extraction schemes formed a homogenous group. As indicated by the calculation, this was obviously not the case. While there was no statistically significant difference between the RTP assay and the Qiagen assay, significant evidence for later Ct values after nucleic acid extraction with the MagaBio assay compared to both competitor assays could be shown, indicating slightly reduced sensitivity by a mean of less than 3 Ct steps, corresponding to less than a decadic logarithmic step. This phenomenon is even more pronounced by the fact that due to a higher extracted sample volume with a proportionally lower eluate volume compared to the competitor assays, even a slightly higher concentration of *T. cruzi* DNA in the eluates after MagaBio assay-based extraction might have been expected. This observation suggests that the MagaBio assays either resulted in a lower yield of target DNA in the eluates or, alternatively, were associated with a chemical component within the eluates that interfered with the amplification efficiency of the *T. cruzi*-specific PCR.

Although there are considerably more nucleic acid extraction assays available on the market that might have been included in the assessment, the included assays were not arbitrarily chosen. The RPT assay had already been applied for the nucleic acid extractions in the course of the initial study on the evaluation of the applied *T. cruzi*-specific PCR [15] and is the standard procedure included in the diagnostic workflow at the participating laboratory in Würzburg, Germany. The MagaBio assay is the standard nucleic acid extraction scheme applied by the Colombian project partners for the *T. cruzi* PCRs on site in the area of endemicity. The Qiagen assay is the standard approach for nucleic acid extraction from serum samples in the participating laboratory in Hamburg, Germany, and has already been successfully applied for the diagnosis of visceral leishmaniasis from human serum samples by the German National Reference Center for Tropical Pathogens [26]. Accordingly, the choice of the assays reflects the diagnostic real-life situation in settings where the molecular diagnosis of *T. cruzi* and other parasites from patient sera is indeed applied.

The study has a number of limitations. First, only a limited number of pre-characterized *T. cruzi*-positive reference materials were available for the study. However, considering the good pre-characterization of the reference material, it remains difficult to obtain higher quantities even in the course of multicentric assessments as conducted here. Second, the study was performed with stored residual samples, so DNA degradation in comparison to the time of sample acquisition cannot be excluded despite frozen storage. To circumvent this problem, all three extraction schemes were applied in close temporal association to keep DNA degradation-associated bias as low as possible for the comparison. So, comparable sample conditions can be assumed for all extraction approaches. Third, nucleic acid quantification within the eluates was not performed. The diagnostic workflow in the study mimicked the workflow as performed in the diagnostic routine, in which such a step would not be part of the standard procedures. So, the PCR results were chosen as the only outcome parameter. Fourth, no extraction assay providing particularly harsh extraction conditions especially adapted to parasites [8,19,20,21,22,23,24] was included in the assessment, as the amplification of freely circulating DNA was intended. As stated above, the choice of the assessed assays was meant to reflect the diagnostic real-life conditions at the participating study sites.

## 5. Conclusions

The study confirmed acceptable diagnostic sensitivity >95%, as calculated for the tested reference materials, with all compared nucleic acid extraction assays. Focusing on the qualitative results, the ranking order of diagnostic reliability was RTP assay > MagaBio assay > Qiagen assay. Focusing on the comparison of the cycle threshold (Ct) values, the ranking order was RTP assay = Qiagen assay > MagaBio assay. These rankings are associated with a residual uncertainty arising from the fact that limited amounts of residual sample materials did not allow repeated nucleic acid extractions with each assay from the same sample. So, confirmation by testing in replicas was unfeasible. However, even minor differences may be of relevance for individual diagnostic decisions in the case of DNA quantities in serum close to the diagnostic detection threshold [15]. The major strength of the study, despite its abovementioned limitations, is the comparably high number of well-characterized residual sample materials available for the assessment. Accordingly, it can be assumed that the conclusions of the study are empirically well confirmed.

## Figures and Tables

**Figure 1 microorganisms-10-01554-f001:**
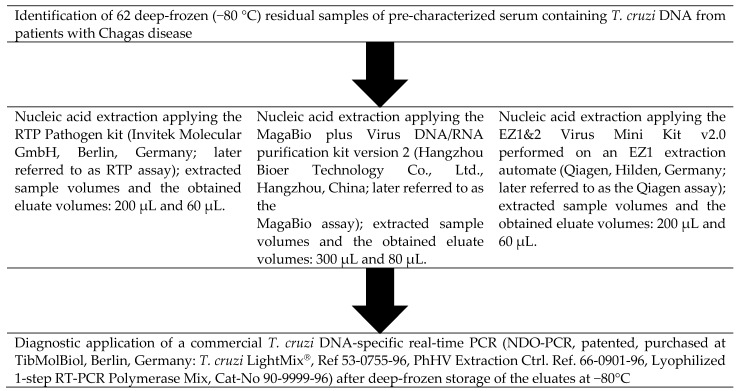
Flow-chart of the diagnostic workflow.

**Table 1 microorganisms-10-01554-t001:** Mean values and standard deviations of the cycle threshold (Ct) values measured after nucleic acid extraction with the different assays with the 59 samples, which showed positive *T. cruzi*-specific PCR results with eluates from all three assessed nucleic acid extraction schemes.

	Nucleic Acid Extraction with the RTP Assay	Nucleic Acid Extraction with the MagaBio Assay	Nucleic Acid Extraction with the Qiagen Assay
Mean value of the measured Ct values	30.8	33.4	31.8
Standard deviation of the measured Ct value	3.0	2.8	3.6

## Data Availability

All relevant data are provided in the manuscript. Raw data can be provided at reasonable request.

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
