# Peer review of "Comparison of the Influence of Different Nucleic Acid Extraction Assays on the Sensitivity of Trypanosoma cruzi-Specific Real-Time PCR"

_microorganisms, 2022, doi:10.3390/microorganisms10081554_

Round 1
Reviewer 1 Report
The article „Comparison of the Influence of Different Nucleic Acid Extraction Assays on the Sensitivity of Trypanosoma cruzi-Specific Real-Time PCR” focuses on describing the results from the comparison of three commercial extraction kits for identification via real time PCR DNA of T. cruzi.
The M&M are performed in clear and easy to read way.
In my opinion the article is well written, the tittle is accurate, the abstacrt provide a precise summary and the structure is well defined. However, it would be beneficial if the authors added the results from the DNA concentration measurement if they have them.
In summary, I consider this work to be of high quality and apart from this one suggestion, I have nothing to add.
Author Response
The article „Comparison of the Influence of Different Nucleic Acid Extraction Assays on the Sensitivity of Trypanosoma cruzi-Specific Real-Time PCR” focuses on describing the results from the comparison of three commercial extraction kits for identification via real time PCR DNA of T. cruzi.
The M&M are performed in clear and easy to read way.
In my opinion the article is well written, the tittle is accurate, the abstacrt provide a precise summary and the structure is well defined. However, it would be beneficial if the authors added the results from the DNA concentration measurement if they have them.
In summary, I consider this work to be of high quality and apart from this one suggestion, I have nothing to add.
As already stated as the third limitation of the limitations paragraph at the end of the discussion, quantification data are, unfortunately, indeed not available. The study was planned to reflect diagnostic routine-like conditions and in none of the participating affiliations, DNA-measurement prior to diagnostic PCR is a diagnostic standard procedure.
Reviewer 2 Report
The manuscript entitled: “Comparison of the Influence of Different Nucleic Acid Extraction Assays on the Sensitivity of Trypanosoma cruzi-Specific Real-Time PCR” aimed to optimize the nucleic acid extraction for Changas disease in order to improve the sensitivity for T. cruzi detection from human serum samples. Currently, Changas disease can not be diagnosed by a molecular test with high accuracy, so new nucleic acid extraction techniques are investigated for a more optimized diagnostic process. The manuscript is well written, but some issues should be clarified before acceptance for publication.
1. Define all the abbreviations at the first use in the text, eg: PCR, DNA, NDO, RTP, etc.
2. In material and methods a schematic representation of the study design will help the reader to better understand the study design
3. The authors clearly presented the limitations of the study, they have also to include the strengths.
Author Response
The manuscript entitled: “Comparison of the Influence of Different Nucleic Acid Extraction Assays on the Sensitivity of Trypanosoma cruzi-Specific Real-Time PCR” aimed to optimize the nucleic acid extraction for Changas disease in order to improve the sensitivity for T. cruzi detection from human serum samples. Currently, Changas disease can not be diagnosed by a molecular test with high accuracy, so new nucleic acid extraction techniques are investigated for a more optimized diagnostic process. The manuscript is well written, but some issues should be clarified before acceptance for publication.
Define all the abbreviations at the first use in the text, eg: PCR, DNA, NDO, RTP, etc.
Authors: Thank you for your response. Actually there is a molecular test with high accuracy, sensitivity and specificity as published by Kann et al 2020 and 2022, it is the NDO-real time PCR.
- Kann S., Kunz M., Hansen J., Sievertsen J., Crespo J.J., Loperena A., Arriens S., Dandekar T., Chagas Disease: Detection of Trypanosoma cruzi by a New, High-Specific Real Time PCR, J. Clin. Med., 2020, 5, 1517; doi:10.3390/ jcm 9051517
- Kann S., Dib J.C., Aristizabal A., Concha Mendzos G., Soto Lacouture H.D., Hartmann M., Frickmann H., Kreienbrock L., Diagnosis and Prevalence of Chagas Disease in an Indigenous Population of Colombia, Microorganisms 2022,10,1427, https://doi.org/10.3390/microorganisms10071427
As requested, the abbreviations used in the manuscript have been spelled out, thank you for that hint.
In material and methods a schematic representation of the study design will help the reader to better understand the study design.
As requested, a respective flow-chart has been added to the methods chapter as a new figure 1.
The authors clearly presented the limitations of the study, they have also to include the strengths.
As requested, we have highlighted the strengths of the study in the conclusions paragraph. In
particular, we have now mentioned the rare availability of well characterized residual-sample material, which made the here-presented assessment feasible.
Reviewer 3 Report
Chagas disease is one of the most prevalent neglected tropical diseases, more than 6 million people are infected, more than 73 million people are at risk of infection and therefore optimising their diagnosis is crucial.
The manuscript is well written, the tittle and the introduction are accurate, and the structure is well defined. However, I consider the assays should be better conducted to support the objective of the manuscript and the conclusions are not supported by the results. I explain my concerns in detail below.
Major comments:
- Abstract provide a precise summary however, I do not consider that the last sentence of the abstract (lines 31-32) is supported by the results of this manuscript. The same sentence appears in the conclusion section. Regarding this, I strongly encourage the authors to conduct assays with less amount of circulating parasites DNA to see differences or optimisation strategies for the different extraction kits used.
- How many times was the extraction repeated with each method for the different samples? In the conclusions section, the authors rank the diagnostic reliability of the different extraction methods, but this cannot be confirmed if there are no replicates of the extracted samples.
Minor comments:
- Page 2, Line 82. Missing bracket
- Page 2, Line 84. Missing symbol plus/minus
Author Response
Chagas disease is one of the most prevalent neglected tropical diseases, more than 6 million people are infected, more than 73 million people are at risk of infection and therefore optimising their diagnosis is crucial.
The manuscript is well written, the tittle and the introduction are accurate, and the structure is well defined. However, I consider the assays should be better conducted to support the objective of the manuscript and the conclusions are not supported by the results. I explain my concerns in detail below.
Major comments:
- Abstract provide a precise summary however, I do not consider that the last sentence of the abstract (lines 31-32) is supported by the results of this manuscript. The same sentence appears in the conclusion section. Regarding this, I strongly encourage the authors to conduct assays with less amount of circulating parasites DNA to see differences or optimisation strategies for the different extraction kits used.
We agree that other pathogens with less circulating DNA in serum might have led to more interpretable results. Accordingly, we have removed the respective sentence from the abstract and from the conclusions.
- How many times was the extraction repeated with each method for the different samples? In the conclusions section, the authors rank the diagnostic reliability of the different extraction methods, but this cannot be confirmed if there are no replicates of the extracted samples.
In line with your suggestion, we have admitted in the conclusions that the rankings are associated with a residual uncertainty arising from the fact that limited amounts of residual sample materials did not allow repeated nucleic acid extractions with each assay from the same sample. So, confirmation by testing in replicas was unfeasible.
Minor comments:
- Page 2, Line 82. Missing bracket
- Page 2, Line 84. Missing symbol plus/minus
Both mistakes were corrected.
Round 2
Reviewer 2 Report
The authors addressed all my comments. The manuscript is ready for acceptance.
Reviewer 3 Report
Although I believe it would be of great value to the study to have replicates of the different extraction methods to make this study much more interesting and highly applicable in the field, I understand that the availability of biological samples is limited. The authors have modified the conclusions of the study according to their results/conditions of the work they propose.